# Adequacy of using a single nasal swab for rapid influenza diagnostic testing, PCR, and whole genome sequencing

Jonathan L. Temte[1], Cristalyne Bell[1]*, Maureen D. Goss[1], Erik Reisdorf[2], John Tamerius[3], Sushruth Reddy[3], Richard Griesser[2], Shari Barlow[1], Emily Temte[1], Mary Wedig[2], Peter A. Shult[2]

1 Department of Family Medicine and Community Health, School of Medicine and Public Health, University of Wisconsin, Madison, Wisconsin, United States of America, 2 Wisconsin State Laboratory of Hygiene, Madison, Wisconsin, United States of America, 3 Quidel Corporation, San Diego, California, United States of America

* cristalyne.bell@fammed.wisc.edu

**Data Availability Statement:** Data were submitted to Harvard Dataverse: https://doi.org/10.7910/DVN/RLXPUM.

## Abstract

Rapid influenza diagnostic tests (RIDT) demonstrate varying sensitivities, often necessitating reverse transcriptase polymerase chain reaction (RT-PCR) to confirm results. The two methods generally require separate specimens. Using the same anterior nasal swab for both RIDT and molecular confirmation would reduce cost and waste and increase patient comfort. The aim of this study was to determine if RIDT residual nasal swab (rNS) specimens are adequate for RT-PCR and whole genome sequencing (WGS). We performed RT-PCR and WGS on paired rNS and nasopharyngeal or oropharyngeal (NP/OP) swab specimens that were collected from primary care patients across all ages. We randomly selected 199 and 40 paired specimens for RT-PCR and WGS, respectively, from the 962 paired surveillance specimens collected during the 2014–2015 influenza season. Sensitivity and specificity for rNS specimens were 81.3% and 96.7%, respectively, as compared to NP/OP specimens. The mean cycle threshold (Ct) value for the NP/OP specimen was significantly lower when the paired specimens were both positive than when the NP/OP swab was positive and the nasal swab was negative (25.5 vs 29.5; p<0.001). Genomic information was extracted from all 40 rNS specimens and 37 of the 40 NP/OP specimens. Complete WGS reads were available for 67.5% (14 influenza A; 13 influenza B) of the rNS specimens and 59.5% (14 influenza A; 8 influenza B) of the NP/OP specimens. It is feasible to use a single anterior nasal swab for RIDT followed by RT-PCR and/or WGS. This approach may be appropriate in situations where training and supplies are limited. Additional studies are needed to determine if residual nasal swabs from other rapid diagnostic tests produce similar results.

## Introduction

Rapid antigen tests are increasingly common in clinical and community settings. They are easy to use, relatively inexpensive, and provide quick results that allow for near-real time

**Funding:** JLT received funding from Quidel Corporation to test 200 banked residual nasal swab specimens at the WSLH and has previously received additional in-kind-support. The surveillance data and specimens were originally collected as part of the Influenza Incidence Surveillance Project (IISP). The Wisconsin Department of Health Services is the primary recipient of the IISP grant and distributes funding to JLT to coordinate the program for Wisconsin. JT and SR are employed by and receive a salary from Quidel Corporation, and they provided feedback on this manuscript prior to submission. No other funders contributed to this manuscript.

**Competing interests:** I have read the journal's policy and the authors of this manuscript have the following competing interests: Quidel Corporation employs John Tamerius and Sushruth Reddy. Jonathan Temte has received in-kind material support and financial support from Quidel Corporation. No other authors have competing interests to declare.

decision-making. Due to a broad range of sensitivity, and in certain circumstances, it may be necessary to confirm rapid test results through reverse transcriptase polymerase chain reaction (RT-PCR) [1, 2]. Moreover, additional evaluation via whole genome sequencing (WGS) can provide important disease surveillance information. Traditionally, rapid and molecular testing required two separate specimens—an anterior nasal swab and a nasopharyngeal or oropharyngeal (NP/OP) swab [3, 4].

A NP/OP swab is still considered the gold standard for RT-PCR and WGS, but anterior nasal swabs have become more common [5–7]. Using a single anterior nasal swab for rapid testing, RT-PCR, and/or WGS could decrease medical waste and supply costs and increase patient comfort. The efficacy of using a residual nasal swab (rNS) for molecular testing, following preparation for and performance of rapid testing, is not well understood, but a couple studies have showed that a rNS may be sufficient for performing RT-PCR and WGS on prospective SARS-CoV-2 specimens [8, 9]. To determine whether a single anterior nasal swab is sufficient for confirming rapid influenza diagnostic test (RIDT) results, we performed RT-PCR on paired NP/OP and rNS specimens collected from patients of all ages in primary care clinics. Additionally, we performed WGS on a subset of paired specimens.

## Methods

### Setting and population

Data and specimens were collected through the Wisconsin Influenza Incidence Surveillance Project (W-IISP) during the 2014–2015 influenza season. The surveillance program has been described elsewhere [3]. Briefly, W-IISP includes five primary care clinics located in urban (2), suburban (1), and rural (2) areas within Dane County, Wisconsin. Four of the clinics are University of Wisconsin family medicine residency clinics. Patients of all ages were eligible for inclusion if the clinician identified the presence of an acute respiratory illness and the patient had at least two acute respiratory tract symptoms (nasal discharge, nasal congestion, sore throat, cough, fever) that began within seven days of their clinic visit.

### Sample collection and preparation

Clinicians collected an anterior nasal specimen by inserting a Puritan Sterile Foam Tipped Applicator one inch into one of the patient's nostrils, rotating the swab three times, and returning the swab to the paper sheath. Clinicians subsequently obtained NP/OP specimens from the nasopharynx or, more commonly, high posterior pharynx using a Copan FLOQS-wabs flocked swab. The NP/OP swab was sealed in 3.0 ml Remel MicroTest M4RT viral transport medium (VTM) and stored at 2–8˚C.

### Diagnostics

RIDT was performed at the clinic laboratory. The nasal swab specimen was handled and processed according to the Quidel Sofia Influenza A+B Fluorescent Immunoassay (FIA) package insert [10]. As directed, the nasal swab was inserted into a test vial containing added lysis buffer. After the completion of the RIDT, the rNS and any unused lysis buffer were sealed in a 3.0 ml Remel MicroTest M4RT VTM tube. The rNS tube was then packaged with the corresponding NP/OP VTM tube and both were shipped via courier to the Wisconsin State Lab of Hygiene (WSLH), generally within 24 hours of sample collection.

At the WSLH, an aliquot from the rNS VTM specimen was extracted and stored at -70˚C until the end of influenza season. RT-PCR was performed on NP/OP swabs within 1 to 9 days

of specimen collection (mean = 3.03 days; median = 3 days). NP/OP specimens and thawed rNS specimens were tested for the presence of influenza A and B using the in-vitro diagnostic (IVD) approved CDC Human Influenza Virus Real-time RT-PCR Diagnostic Panel (Cat.# FluIVD03) [11]. Influenza A and B positive specimens were subtyped using the same testing kit. Next-generation whole-genome sequencing was performed using multisegment reverse transcription-PCR (M-RTPCR) with Invitrogen Super-Script III One-Step RT-PCR with Platinum Taq High Fidelity enzyme (Thermo Fisher Scientific Inc.) to simultaneously amplify eight genomic RNA segments of influenza A and B viruses [12, 13]. Indexed sequencing libraries were generated from the M-RTPCR amplicons using the Nextera XT Sample Preparation Kit (Illumina, San Diego, CA, USA) and sequenced on the MiSeq platform using MiSeq Reagent Kit v2 (300-cycles) reagents [14]. Raw sequencing reads were analyzed using the Iterative Refinement Meta-Assembler (IRMA) approach and consensus sequences were manually checked to verify data quality [15].

## Subsets of specimens

For assessment of comparability, paired NP/OP and rNS specimens from the 2014–2015 influenza season were grouped based on the NP/OP influenza RT-PCR result (negative, influenza A positive, influenza B positive). Residual specimens were then randomly drawn from each group [influenza A (n = 80); influenza B (n = 60); negative (n = 60)] for validation testing. One randomly selected rNS specimen (influenza A) was not tested due to insufficient specimen volume, leaving 199 paired specimens for RT-PCR comparison.

We subsequently selected 40 paired specimens (20 influenza A and 20 influenza B) for WGS. To do this, we removed 86 paired specimens from the original 199 that had at least one negative RT-PCR result (rNS and/or NP/OP). From the remaining 113 pairs, we removed 49 additional specimen pairs for which the rNS cycle threshold (Ct) value was ≥29.0. This left 34 influenza A specimen pairs and 30 influenza B specimen pairs from which we randomly selected a total of 80 specimens (40 paired rNS and NP/OP specimens).

## Statistical analyses

Characteristics of the total pool and the random samples of influenza A, influenza B, and influenza negative sets were compared for proportion of cases meeting influenza-like illness (ILI) criteria, sex ratio, and clinician-rated severity on a 3-point scale using a Chi-square test. ILI was defined as the presence of fever and cough and/or sore throat for cases aged 2 years or older, and fever with any respiratory symptom for cases less than 2 years. Severity was rated as mild, moderate, or severe based on clinician assessment. A Kruskal-Wallis test was used to compare time in days from illness onset to specimen collection.

Sensitivity, specificity, positive predictive value, negative predictive value, and accuracy of the nasal swab specimens were calculated using standard methods, where the NP/OP swab was used as the 'gold standard' comparator. Confidence intervals were calculated using standard methods. The mean Ct values for each group were compared through a paired t-test.

We assessed the performance characteristics for influenza A [(n = 79) versus influenza B plus influenza negative (n = 120)], influenza B [(n = 60) versus influenza A plus influenza negative (n = 139)], and influenza A or B [(n = 139) versus influenza negative (n = 60)]. Due to the possible effects of time on RT-PCR result for the archived nNS specimens, we assessed the difference in Ct values between NP/OP and rNS specimens as a function of specimen collection date using the Pearson correlation. Separate analyses were performed for influenza A and influenza B specimens.

### Ethical approval

Anonymized samples were tested and ethical approval was not required. Data and specimens were collected in a public health surveillance program which has been deemed exempt by the University of Wisconsin Health Sciences Institutional Review Board.

## Results

Between July 1, 2014 and June 30, 2015, 962 paired specimens were collected from 986 patient encounters (97.6% dual swab compliance). Nasal swabs were not collected for 21 cases; NP/OP swabs were not collected for 3 cases (Fig 1). RT-PCR detected 248 (25.8%) cases of influenza—166 (17.3%) influenza A (H3N2), 4 (0.4%) unsubtypeable influenza A, and 78 (8.1%) influenza B. The unsubtypeable specimens were removed from further analyses. The remaining 958 specimens were separated into three groups from which specimens were randomly selected using the random sample application in Minitab to produce a set of 200 specimens for validation testing (Table 1).

The proportions of cases meeting influenza-like illness criteria, sex ratios, mean severity levels, ages, time from illness onset to specimen collection, and Ct values of the specimen pools were comparable to proportions found in the randomly selected samples (Table 2). No significant differences were noted in the demographic and clinical characteristics between the total pool of specimens and the random sample of specimens for each outcome group.

In the random sample, RT-PCR detected influenza in 113 of 139 rNS specimens for which the corresponding NP/OP specimens were positive for influenza, resulting in an estimated overall sensitivity of 81.3% (95% CI: 73.8–87.4). The RT-PCR results for 2 rNS specimens were inconclusive and counted as false negatives. Influenza was detected in 2 nasal swab specimens for which the corresponding NP/OP specimens were negative for influenza. Although the negative NP/OP results may have been due to discrepancies in swabbing technique and/or obtaining an insufficient specimen, we categorized the rNS results as false positive results. The specificity was estimated to be 96.7% (88.5–99.6). The performance characteristics of rNS, as compared to NP/OP swabs, for identification of influenza A and influenza B are displayed in Table 3.

Of the 113 rNS specimens that tested positive for influenza with a corresponding positive NP/OP specimen, 32 (28%) had a lower Ct value. For paired specimens in which both the NP/OP and the nasal swab were positive, the mean Ct value for RT-PCR for the NP/OP specimens was significantly lower than for those in which the NP/OP was positive and the rNS specimen was negative for influenza (25.5 vs 29.5; p<0.001).

Although nasal swab aliquots were frozen at -70°C for longer and variable periods and the NP/OP specimens were tested as they were received, it is unlikely this time affected the quality of the nasal swab specimens. Length of freezing time was not correlated with differences in Ct values between rNS and NP/OP specimens for influenza A (p = 0.598) or influenza B (p = 0.426).

In the subset of the 40 paired specimens, three NP/OP specimens were excluded due to insufficient volume. For the remaining 77 specimens, 27 of the 40 rNS specimens (67.5%; 14 influenza A and 13 influenza B) and 22 of the 37 NP/OP specimens (59.5%; 14 influenza A and 8 influenza B) produced complete genomic information. To be considered complete, all segments needed to be assembled (8/8 influenza genes) and pass a quality control test. For the 28 incomplete specimens, 16 were unable to assemble 1 segment, 7 were unable to assemble 2 segments, and 5 were unable to assemble 3 or more segments.

Specimens with a complete genome had a mean Ct of 23.38 (+/- 3.00) and specimens with an incomplete genome had a mean Ct of 26.36 (+/- 4.26; t = -3.58; p <0.001). The mean Ct values for the 20 influenza A and 20 influenza B rNS specimens were 24.75 [range: 21.08–28.67]

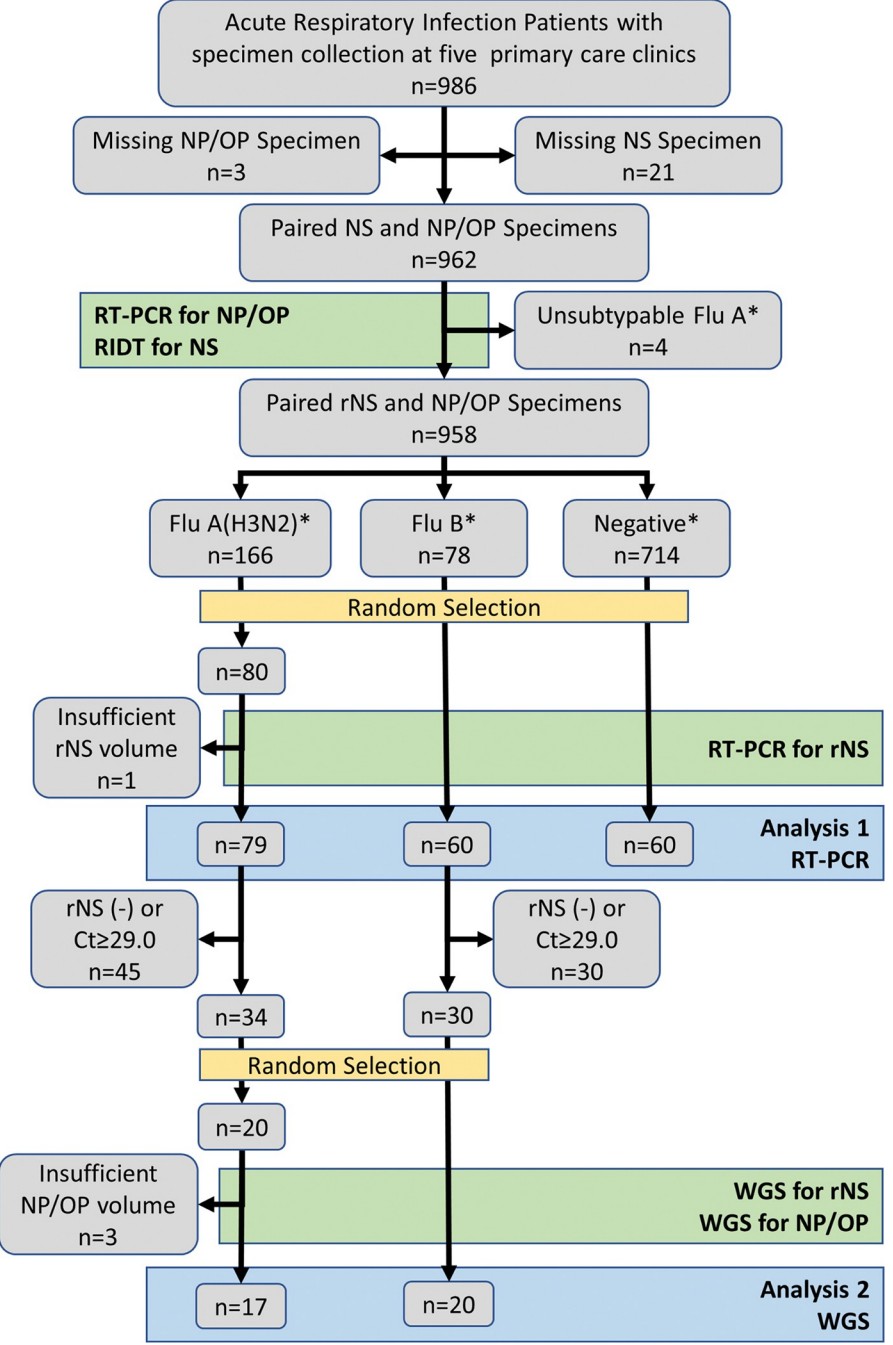

**Fig 1. Flow diagram showing selection of specimens for testing and comparison starting with 986 primary care patients with acute respiratory infection.** NP/OP = nasopharyngeal/oropharyngeal specimen; NS = nasal swab specimen; RT-PCR = reverse transcription polymerase chain reaction; RIDT = rapid influenza diagnostic test; rNS = residual NS specimen following RIDT; Ct = cycle threshold value from RT-PCR; WGS = whole genome sequencing. Laboratory testing is indicated by green boxes; random selection of specimens is indicated by yellow boxes; comparative analyses is indicated by blue boxes.

and 24.76 [range: 19.04–28.59], respectively. The mean Ct values were similar between the 40 rNS specimens and 37 NP/OP specimens (24.8 vs. 24.7). The differences in mean Ct values were not statistically significant regardless of virus or swab type.

**Table 1. Available paired residual nasal swab (rNS) and nasopharyngeal/oropharyngeal (NP/OP) swabs surveillance specimens with subsets randomly selected for analysis.** Dane County, Wisconsin; July 1, 2014 –June 30, 2015.

|  | Influenza A (H3N2) | Influenza A unsubtypeable | Influenza B | Negative | Total |
|---|---|---|---|---|---|
| Total Specimens | 166 (17.3%) | 4 (0.4%) | 78 (8.1%) | 714 (74.2%) | 962 |
| Randomly-Selected Specimens | 80* | 0 | 60 | 60 | 200 |

*One randomly selected, archived specimen did not have sufficient material for PCR testing.

**Table 2. Demographic and clinical information for the total population and those randomly selected for analysis.**

| Case Characteristic | Influenza A | | Influenza B | | Negative | |
|---|---|---|---|---|---|---|
| | Total pool N = 166 | Sample N = 79 | Total pool N = 78 | Sample N = 60 | Total pool N = 714 | Sample N = 60 |
| Age in years (mean) [range] | 36.2 | 35.2 [0–86] | 41.9 | 43.3 [0–74] | 35.2 | 35.6 [0–82] |
| Female (%) | 56.0 | 58.8 | 51.3 | 46.6 | 61.5 | 63.3 |
| Days from illness onset (mean) [range] | 3.1 | 3.0 [0–8] | 3.7 | 3.5 [0–14] | 4.1 | 4.1 [1–10] |
| Severity (mean) | 1.75 | 1.75 | 1.68 | 1.65 | 1.72 | 1.75 |
| ILI (%) | 70.5 | 63.8 | 67.9 | 70.0 | 50.0 | 48.3 |
| Ct value (mean) [range] | 26.7 | 26.8 [17.5–36.7] | 25.8 | 25.7 [16.7–34.5] | N/A | N/A |

**Table 3. Performance characteristics of residual nasal swab (rNS) specimens as compared to nasopharyngeal or high oropharyngeal (NP/OP) for RT-PCR.**

|  | Sensitivity | Specificity | Positive predictive value(PPV) | Negative predictive value(NPV) | Accuracy |
|---|---|---|---|---|---|
| Combined Influenza A and B | 81.3% (73.8–87.4) | 96.7% (88.5–99.6) | 98.3% (93.5–99.6) | 69.1% (61.1–76.0) | 85.9% (80.3–90.4) |
| Influenza A | 79.8% (69.2–88.0) | 98.5% (94.6–99.8) | 96.9% (88.8–99.2) | 88.9% (83.8–92.5) | 85.6% (86.7–94.8) |
| Influenza B | 86.7% (75.4–94.1) | 100% (97.4–100) | 100% | 94.6% (90.1–97.1) | 96.0% (92.2–98.3) |

## Discussion

When we compared results from paired anterior nasal and NP/OP specimens, we found it was feasible to perform RT-PCR and WGS on a rNS after the specimen had been processed for RIDT. There was no significant difference in the mean Ct values for rNS and NP/OP specimens, suggesting that specimens collected from the anterior passage of a nose can contain adequate amounts of viral material—sometimes even surpassing a more invasive NP/OP swab. Moreover, we found that rNS specimens were not inferior to NP/OP specimens for WGS. Although the subset of paired specimens had comparable mean Ct values, complete genomes were assembled in more rNS specimens than NP/OP specimens.

The rNS demonstrated high specificity (>96% overall) and moderate sensitivity (>80%). Our specificity estimates are conservative because we treated 2 RT-PCR-positive rNS specimens as false positives, but they may have represented false-negative NP/OP specimens. The reduced sensitivity may have been due to sample collection technique, dilution of the specimen following processing for RIDT, and/or a delay in testing for the rNS. However, the fact that it was a residual swab instead of an unused nasal swab didn't seem to matter as our results are comparable to the existing literature. In another study that compared the accuracy and discomfort of different swabs, nasal swabs collected for RT-PCR were 99.1% specific and 84.4% sensitive [16].

Following the SARS-CoV-2 pandemic, anterior nasal swabs and rapid antigen tests have become more widely used for diagnostics and disease surveillance. Although an NP/OP swab is still the gold standard, it is not always feasible to collect a secondary swab, particularly when

there are supply chain constraints, as was seen early in the pandemic. Using a rNS for RT-PCR and WGS has several advantages, but research on the efficacy is still limited. In 2016, we initiated a randomized controlled study in 20 long-term care facilities where nurses and nursing assistants were able to collect a single nasal swab for both RIDT and RT-PCR [17]. The rNS specimens were used successfully for confirmation of influenza and detection of additional viruses by RT-PCR. More recently, a study showed that the residual buffer left over from a rapid SARS-CoV-2 test could be used to confirm a positive result through RT-PCR [9]. Another study demonstrated that WGS could be performed on both residual nasal swabs and the positive line on a lateral flow assay [8].

## Limitations

Our results were based on the influenza strains that were prevalent in southcentral Wisconsin during the 2014–2015 influenza season. Influenza subtypes circulating during this period were limited to the H3N2 strain for influenza A and primarily the Yamagata strain for influenza B (92%). Further, our results were based on rNS specimens from one RIDT and may vary depending on the type of test or the virus being detected. Our specimens were extracted from only one nostril, but the current guidance is to insert the anterior nasal swab into both nostrils [18].

## Conclusion

Residual nasal swabs that were used for RIDT performed well when they were subsequently used for RT-PCR and WGS. Our results suggest that using rNS specimens may be a possible alternative to collecting a secondary NP/OP swab when additional laboratory testing is necessary. This is especially important for when cost, lack of training, supply shortages, and other barriers interfere with adequate testing and disease surveillance. Additional studies are needed to determine if residual nasal swabs from other rapid diagnostic tests produce similar results.

## Author Contributions

**Conceptualization:** Jonathan L. Temte, Erik Reisdorf, John Tamerius, Sushruth Reddy, Shari Barlow, Emily Temte, Mary Wedig, Peter A. Shult.

**Data curation:** Maureen D. Goss, Erik Reisdorf, John Tamerius, Sushruth Reddy, Shari Barlow, Emily Temte, Mary Wedig, Peter A. Shult.

**Formal analysis:** Jonathan L. Temte, Erik Reisdorf, John Tamerius, Sushruth Reddy, Richard Griesser, Shari Barlow, Emily Temte, Mary Wedig, Peter A. Shult.

**Funding acquisition:** Jonathan L. Temte.

**Methodology:** Jonathan L. Temte.

**Writing – original draft:** Jonathan L. Temte, Cristalyne Bell, Maureen D. Goss.

**Writing – review & editing:** Jonathan L. Temte, Cristalyne Bell, Maureen D. Goss, Erik Reisdorf, John Tamerius, Sushruth Reddy, Richard Griesser, Shari Barlow, Emily Temte, Mary Wedig, Peter A. Shult.

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
