## [Decision Letter · Decision Letter 0]

23 Jan 2023

PGPH-D-22-01919

Adequacy of using a residual nasal swab specimen from a rapid influenza diagnostic test to perform PCR and whole genome sequencing

Dear Dr. Bell,

Thank you for submitting your manuscript to PLOS Global Public Health. After careful consideration, we feel that it has merit but does not fully meet PLOS Global Public Health’s publication criteria as it currently stands. Therefore, we invite you to submit a revised version of the manuscript that addresses the points raised during the review process.

We look forward to receiving your revised manuscript.

Kind regards,

Julio Croda, Ph.D, M.D.

Academic Editor

Journal Requirements:

1. Please send a completed 'Competing Interests' statement, including any COIs declared by your co-authors. If you have no competing interests to declare, please state "The authors have declared that no competing interests exist". Otherwise please declare all competing interests beginning with the statement "I have read the journal's policy and the authors of this manuscript have the following competing interests:"

3. We do not publish any copyright or trademark symbols that usually accompany proprietary names, eg (R), (C), or TM  (e.g. next to drug or reagent names). Please remove all instances of trademark/copyright symbols throughout the text, including R and TM on page 8 and 9.

4. In the online submission form, you indicated that your data will be submitted to a repository upon acceptance.  We strongly recommend all authors deposit their data before acceptance, as the process can be lengthy and hold up publication timelines. Please note that, though access restrictions are acceptable now, your entire data will need to be made freely accessible if your manuscript is accepted for publication. This policy applies to all data except where public deposition would breach compliance with the protocol approved by your research ethics board. If you are unable to adhere to our open data policy, please kindly revise your statement to explain your reasoning and we will seek the editor's input on an exemption. Please be assured that, once you have provided your new statement, the assessment of your exemption will not hold up the peer review process.

Additional Editor Comments (if provided):

Reviewers' comments:

Reviewer's Responses to Questions

**Comments to the Author**

1. Does this manuscript meet PLOS Global Public Health’s publication criteria? Is the manuscript technically sound, and do the data support the conclusions? The manuscript must describe methodologically and ethically rigorous research with conclusions that are appropriately drawn based on the data presented.

Reviewer #1: No

Reviewer #2: Partly

2. Has the statistical analysis been performed appropriately and rigorously?

Reviewer #1: No

Reviewer #2: I don't know

3. Have the authors made all data underlying the findings in their manuscript fully available (please refer to the Data Availability Statement at the start of the manuscript PDF file)?

Reviewer #1: No

Reviewer #2: Yes

4. Is the manuscript presented in an intelligible fashion and written in standard English?

Reviewer #1: Yes

Reviewer #2: Yes

5. Review Comments to the Author

Reviewer #1: The authors measured the accuracy of rapid influenza diagnostic testing of anterior nasal swabs compared to RT-PCR from nasopharyngeal or oropharyngeal (NP/OP) swab specimens. They selected 199 participants from a previous influenza prospective surveillance study and measured the accuracy of rapid diagnostic tests using NP/OP RT-PCR results as the gold standard of positivity. This is an important applied public health question. I have several questions about the methods, in particular about WGS methods and results, that should be addressed.

• Abstract. Please define what is meant by “successful” WGS. Is this based on a pre-specified coverage depth and breadth threshold?

• Methods. Please also include a section on whole genome sequencing, including library preparation, sequencing, and bioinformatic analysis.

• Methods. How did recruitment work? Was there informed consent and IRB approval? Was this study exempt?

• Results. Please expand the paragraph on whole genome sequencing. As written, is difficult to understand what is meant by complete WGS reads and what is meant to “WGS was successfully performed.”

• Results. Was Ct associated with coverage? Did coverage differ between viral types?

• Results. Were the WGS data deposited to a publicly available sequence repository?

• Discussion. I’d suggest including that the rNS and NP/OP Cts were not significantly different (if that is the case), rather than the percentage of rNS with lower Ct values.

Reviewer #2: Minor revisions:

1. Tables need legends.

2. Did not have any exclusion criteria?

3. “RT-PCR was performed on NP/OP swabs within 1 to 9 days of specimen collection (mean = 3.03 days; median = 3 days).” Were the NP and OP samples sent together with the rNS? It was not well described what was done with the NP and OP swabs. It just says RT-PCR was done within 1 to 9 days.

4. “Between July 1, 2014 and June 30, 2015, 962 paired specimens were collected from 986 patient encounters (97.6% dual swab compliance). Nasal swabs were not collected for 21 cases; NP/OP swabs were not collected for 3 cases. RT-PCR detected 248 (25.8%) cases of influenza—166 (17.3%) influenza A (H3N2), 4 (0.4%) unsubtypeable influenza A, and 78 (8.1%) influenza B. The unsubtypeable specimens were removed from further analyses. The remaining 958 specimens were separated into three groups from which specimens were randomly selected using the random sample application in Minitab to produce a set of 200 specimens for validation testing (Table 1).” It was not evident during the reading that four of the 962 samples were removed, leaving 958. Perhaps remove these 4 from the table, put an asterisk in local specimens, and put in the legend that they were removed because it was impossible to subtype.

6. PLOS authors have the option to publish the peer review history of their article (what does this mean?). If published, this will include your full peer review and any attached files.

**Do you want your identity to be public for this peer review?** For information about this choice, including consent withdrawal, please see our Privacy Policy.

Reviewer #1: No

Reviewer #2: No

---

## [Decision Letter · Decision Letter 1]

26 Apr 2023

Adequacy of using a single nasal swab for rapid influenza diagnostic testing, PCR, and whole genome sequencing

PGPH-D-22-01919R1

Dear Bell,

We are pleased to inform you that your manuscript 'Adequacy of using a single nasal swab for rapid influenza diagnostic testing, PCR, and whole genome sequencing' has been provisionally accepted for publication in PLOS Global Public Health.

Best regards,

Julio Croda, Ph.D, M.D.

Academic Editor

Reviewer Comments (if any, and for reference):

Reviewer's Responses to Questions

**Comments to the Author**

1. If the authors have adequately addressed your comments raised in a previous round of review and you feel that this manuscript is now acceptable for publication, you may indicate that here to bypass the “Comments to the Author” section, enter your conflict of interest statement in the “Confidential to Editor” section, and submit your "Accept" recommendation.

Reviewer #2: All comments have been addressed

2. Does this manuscript meet PLOS Global Public Health’s publication criteria? Is the manuscript technically sound, and do the data support the conclusions? The manuscript must describe methodologically and ethically rigorous research with conclusions that are appropriately drawn based on the data presented.

Reviewer #2: Yes

3. Has the statistical analysis been performed appropriately and rigorously?

Reviewer #2: I don't know

4. Have the authors made all data underlying the findings in their manuscript fully available (please refer to the Data Availability Statement at the start of the manuscript PDF file)?

Reviewer #2: Yes

5. Is the manuscript presented in an intelligible fashion and written in standard English?

Reviewer #2: Yes

6. Review Comments to the Author

Reviewer #2: (No Response)

7. PLOS authors have the option to publish the peer review history of their article (what does this mean?). If published, this will include your full peer review and any attached files.

**Do you want your identity to be public for this peer review?** For information about this choice, including consent withdrawal, please see our Privacy Policy.

Reviewer #2: No
